# Generalized Laplacian Positional Encoding for Graph Representation Learning

**Sohir Maskey**[*]                                                     MASKEY@MATH.LMU.DE
*Ludwig-Maximilian University of Munich*

**Ali Parviz**[*]                                                       ALI.PARVIZ@MILA.QUEBEC
*MILA, New Jersey Institute of Technology*

**Maximilian Thiessen**                              MAXIMILIAN.THIESSEN@TUWIEN.AC.AT
*TU Wien*

**Hannes Stärk**                                                          HSTARK@MIT.EDU
*Massachusetts Institute of Technology*

**Ylli Sadikaj**                                               YLLI.SADIKAJ@UNIVIE.AC.AT
*University of Vienna*

**Haggai Maron**                                                      HMARON@NVIDIA.COM
*NVIDIA Research*

**Editors:** Sophia Sanborn, Christian Shewmake, Simone Azeglio, Arianna Di Bernardo, Nina Miolane

## Abstract

Graph neural networks (GNNs) are the primary tool for processing graph-structured data. Unfortunately, the most commonly used GNNs, called Message Passing Neural Networks (MPNNs) suffer from several fundamental limitations. To overcome these limitations, recent works have adapted the idea of positional encodings to graph data. This paper draws inspiration from the recent success of Laplacian-based positional encoding and defines a novel family of positional encoding schemes for graphs. We accomplish this by generalizing the optimization problem that defines the Laplace embedding to more general dissimilarity functions rather than the 2-norm used in the original formulation. This family of positional encodings is then instantiated by considering $p$-norms. We discuss a method for calculating these positional encoding schemes, implement it in PyTorch and demonstrate how the resulting positional encoding captures different properties of the graph. Furthermore, we demonstrate that this novel family of positional encodings can improve the expressive power of MPNNs. Lastly, we present preliminary experimental results.

**Keywords:** Graph Neural Networks, Positional Encoding, Graph Laplacian

## 1. Introduction

Over the past few years, graph neural networks have become the most popular tool for processing graph-structured data. Message Passing Neural Networks (MPNNs) (Gilmer et al., 2017), which update node features through local aggregation, are currently the most popular type of GNN architecture. Despite their success across a wide variety of domains, MPNNs have several severe limitations, such as their bounded expressive power (Xu et al., 2018; Morris et al., 2019, 2021) and their inability to solve simple link prediction problems (Zhang et al., 2021). Many recent studies have suggested overcoming these limitations by

---

[*] Equal contribution.

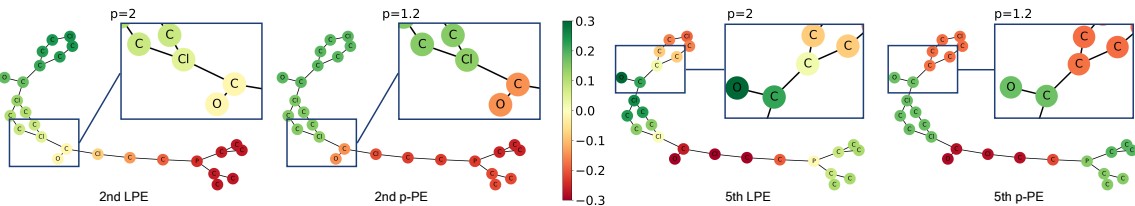

Figure 1: Comparison between $p$-PEs ($p = 1.2$) and LPEs ($p = 2$) on one ZINC-molecule.

using *Positional Encoding* (PE), a node embedding that encodes a node's location in a graph, as its initial feature (Dwivedi et al., 2020). This idea originates from the field of text processing where sequential positional encoding is frequently used when using transformer architectures (Vaswani et al., 2017).

Unfortunately, sequential positional encoding is not suitable for processing graphs since in most cases, nodes in graphs lack a canonical order. One of the most prominent realizations of PE for graphs utilizes the graph Laplacian eigenvectors as node features (Kreuzer et al., 2021; Dwivedi and Bresson, 2020). It has been demonstrated that this PE scheme improves the performance of MPNNs in a variety of tasks (Dwivedi et al., 2020, 2021; Lim et al., 2022). One way to define the Laplacian-based PE (LPE) is as the solution to the minimization problem of finding an embedding $X = [X_{1,:}, \ldots, X_{n,:}] \in \mathbb{R}^{n \times k}$ that adheres to the affinity structure of the graph (Belkin and Niyogi, 2003):

$$\min_{X \in \mathbb{R}^{n \times k}} \sum_{i,j} a_{ij} \|X_{i,:} - X_{j,:}\|_2^2 = \min_X \operatorname{tr}(X^T L X), \quad \text{s.t.} \quad X^T D X = I_k. \tag{1}$$

Here, $A = (a_{i,j})_{i,j=1}^n$ represent the affinities (or adjacency information) between nodes, $D = \operatorname{diag}(A1)$ and $L = D - A$ is the graph Laplacian (Chung, 1997). Minimizing this problem can be accomplished by finding the Laplacian eigendecomposition. Importantly, while the intuition of Equation (1) is clear - finding an embedding whose distance between nodes with high affinity is small - the specific choice of the 2-norm is arbitrary.

The purpose of this paper is to generalize this successful positional encoding scheme. Our main observation is that meaningful node embeddings can be defined as solutions to a more general family of optimization problems, obtained by replacing the 2-norm in Equation (1) with other dissimilarity functions. Motivated by this observation, we first define a novel family of positional encodings based on any dissimilarity function defined in the embedding space. We then instantiate this family by using $p$-norms. We dub the resulting positional encoding scheme *p-PE*. We show how to calculate $p$-PE embeddings using a local minimization scheme and demonstrate that these node embeddings capture different properties of the graph: while LPEs vary smoothly across the graph, when $p \to 1$ $p$-PE behaves like a soft clustering function, and is able to identify meaningful graph structures such as rings in molecules. See Figure 1. We present preliminary experimental results, which indicate that in most cases LPE is still superior to p-PEs. Lastly, we discuss future research directions and challenges encountered in our research on this topic in order to encourage future research on this topic.

## 2. Generalized Laplacian positional encoding

In this subsection, we formulate a novel family of positional encoding schemes for graphs. The main idea is to build on the formulation of the Laplacian positional encoding as a minimizer of the optimization problem in Equation (1) and replace the 2-norm with more general dissimilarity functions. This gives rise to the following optimization:

$$\min_{X \in \mathbb{R}^{n \times k}} \sum_{i,j} a_{ij} d(X_{i,:}, X_{j,:}) \quad \text{s.t.} \quad C(X) = 0, \tag{2}$$

where $d(\cdot, \cdot)$ is a dissimilarity function, $k$ is the dimension of the embedding and $C : \mathbb{R}^{n \times k} \to \mathbb{R}^m$ is a function that formulates the constraints on the embedding matrix $X$, making sure the solution space does not contain any trivial (e.g., constant) solutions[1]. Once computed, $X_{i,:}$ is used as an additional initial features for the $i$-th node.

**Examples.** It is easy to see that when using $d(\cdot, \cdot) = \| \cdot - \cdot \|_2^2$ and an appropriate choice of $C$ we recover the LPE formulation in Equation (1). There are many other choices for $d$ and $C$ which give rise to new, unexplored positional encoding schemes. Let us review several such choices. As for $d$, interesting choices include $p$-norms $d(\cdot, \cdot) = \| \cdot - \cdot \|_p^p$, and clipped or smoothed distance functions. For $C$, besides the normalized orthogonality constraint in Equation (1) one can enforce standard orthogonality, namely $X^T X = I_k$ or constraints that enforce minimal distance between the rows of $X$. The remainder of this paper explores the possibility of using $p$-norms as the distance function, as outlined in the next section.

## 3. $p$-norm based positional encoding

In this section we explore a natural instantiation of Equation (2) based on $p$-norms. Replacing the 2-norm with (normalized) $p$-norms and adding an orthogonality constraint results in the following optimization problem, first introduced by (Luo et al., 2010):

$$\min_{X \in \mathbb{R}^{n \times k}} \sum_k \frac{\sum_{i,j} a_{ij} |X_{i,k} - X_{j,k}|^p}{\|X_{:,k}\|_p^p} \quad \text{s.t.} \quad X^T X = I_k. \tag{3}$$

This formulation is useful for two main reasons: (1) The constraint $X^T X = I_k$ implies that $X$ is constrained to the Stiefel manifold, namely $X \in \text{St}(k, n)$. As a result, optimization can be performed using Riemannian SGD. (2) The solutions to this optimization problem are an approximation of objects known as $p$-eigenvectors, which have appealing theoretical properties. The solutions to this optimization problem are referred to as $p$-PEs.

**Optimization.** As the Stiefel manifold $\text{St}(k, n)$ is a Riemannian manifold we perform Riemannian SGD on $\text{St}(k, n)$ to solve for the minimizer of (3). Riemannian SGD generalizes Euclidean SGD to Riemannian manifolds by replacing the Euclidean gradient by the Riemannian gradient and the Euclidean straight line by geodesics. We implement the Riemannian SGD algorithm, and its variations such as Riemannian Adam, on the Stiefel manifold with geoopt (Kochurov et al., 2020). Minimizing Equation (3) directly for a small value of $p$ often results in convergence to a non-optimal local minimum. Thus we use a

---

1. Note that $d$ can also take $X$ as an input, although this is not reflected in Equation 2 for simplicity.

Table 1: Results on the node classification task.

| model | Cora | Citeseer |
|---|---|---|
| GCN+No PE | $83.6 \pm 0.00$ | $74.7 \pm 0.01$ |
| GCN+1.2-PE | $83.4 \pm 0.00$ | $75.2 \pm 0.01$ |
| GCN+1.6-PE | $83.6 \pm 0.46$ | $73.1 \pm 0.01$ |
| GCN+2-PE | $83.2 \pm 0.00$ | $74.3 \pm 0.01$ |

continuation method, as suggested in (Bühler and Hein, 2009a), that leverages the fact that the minimization problem has an efficient solution for $p = 2$. More precisely, we first solve for the solution of Equation (3) for $p = 2$ using an eigendecomposition of the Laplacian, and then gradually decrease the value of $p$ with the solution at the current $p$ serving as the initialization of the next $p$.

**Relation to $p$-Laplacians and expressive power.** The minimization problem in Equation (3) is tightly connected to a generalization of the Laplacian operator called the $p$-Laplacian (Luo et al., 2010). Remarkably, the solutions to the minimization problem above can be shown to be an approximation of the eigenvectors of this operator (under proper definitions), called $p$-eigenvectors. The $p$-eigenvectors share many properties with the standard eigenvectors (i.e., $p = 2$) of the graph Laplacian. For instance, the number of connected components in the graph is the multiplicity of its first eigenvector. Previous works leveraged $p$-eigenvectors for graph learning: for example, (Bühler and Hein, 2009a) used the fact that for $p \to 1$ the cut obtained by thresholding the first non-trivial $p$-eigenvector converges to the ratio Cheeger cut. Recently, $p$-Laplacians were also used in (Fu et al., 2022) as a motivation for a new GNN architecture. More details on $p$-Laplacians can be found in the appendix. Lastly, previous works have shown that using LPE improves the expressive power of MPNNs (Lim et al., 2022). Next, we generalize this result to $p$-eigenvectors.

**Theorem 1** *For any $p \in (1, \infty)$, MPNNs augmented with at least two $p$-eigenvectors are strictly more expressive than the 1-WL test.*

It is important to note that this statement refers to $p$-eigenvectors rather than $p$-PEs which are an approximation. However, we see in practice that our optimization scheme can produce features that enable us to differentiate 1-WL equivalent graphs.

## 4. Experiments

In this section, we empirically evaluate the quality of $p$-PE on two different popular graph learning tasks: 1) Graph regression on molecular graphs of the ZINC 12K dataset; and 2) Semi-supervised node classification on the Cora and Citeseer datasets. See Appendix C for further details.

In our experiments we consider three settings: 1) No positional encoding, 2) $p$-PE 3) As $p$-PEs have sign ambiguity (like LPE), we randomly flip their signs (flip) and apply SignNet (SN) (Lim et al., 2022) to process them. We focus on two base GNN architectures: GIN with edge features (Xu et al., 2018) and a transformer with sparse attention (Kreuzer et al., 2021). For the baseline, we report the results based on LPEs defined as the eigenvectors of the unnormalized Laplacian (2-PE) and normalized Laplacian (2-PE-Norm.). In all

Table 2: ZINC dataset, 500K parameter budget.

| Model | Test MAE |
|---|---|
| GINE+No PE | $0.23 \pm 0.01$ |
| GINE+2-PE (flip) | $0.231 \pm 0.01$ |
| GINE+2-PE (SN) | $0.249 \pm 0.02$ |
| GINE+2-PE-Norm. (flip) | $0.194 \pm 0.01$ |
| GINE+2-PE-Norm. (SN) | $0.168 \pm 0.02$ |
| GINE+1.2-PE (flip) | $0.205 \pm 0.00$ |
| GINE+1.2-PE (SN) | $0.18 \pm 0.01$ |
| GINE+1.6-PE (flip) | $0.204 \pm 0.00$ |
| GINE+1.6-PE (SN) | $0.171 \pm 0.00$ |
| Transformer+No PE | $0.283 \pm 0.03$ |
| Trans.+2-PE (flip) | $0.351 \pm 0.18$ |
| Trans.+2-PE (SN) | $0.119 \pm 0.00$ |
| Trans.+2-PE-Norm. (flip) | $0.238 \pm 0.02$ |
| Trans.+2-PE-Norm. (SN) | $0.121 \pm 0.00$ |
| Trans.+1.2-PE | $0.296 \pm 0.01$ |
| Trans.+1.6-PE | $0.287 \pm 0.01$ |
| Trans.+1.2-PE (flip) | $0.4 \pm 0.16$ |
| Trans.+1.2-PE (SN) | $0.133 \pm 0.01$ |
| Trans.+1.6-PE (flip) | $0.41 \pm 0.18$ |
| Trans.+1.6-PE (SN) | $0.134 \pm 0.01$ |

experiments we consider $k = 4$. We use the standard train/val/test splits. The results can be found in Table 1-2 and indicate that in most cases LPE is still superior to $p$-PEs.

## 5. Conclusion

We present a novel formulation for graph positional encoding that generalizes Laplacian-based positional encoding. We also provide a concrete implementation of this framework based on $p$-norms. Unfortunately, despite the fact that the proposed positional encoding scheme still appears promising to us, we were not able to demonstrate its effectiveness in the experiments that we conducted.

**Challenges:** The primary challenge of this research direction is to find an efficient and accurate method for minimizing Equation 3. Despite the fact that the optimization scheme we used worked well for $k = 4$ eigenvectors, we found it difficult to optimize for larger values of $k$. Moreover, the optimization time is longer than the time required to solve for LPEs. Developing efficient solvers for these problems is thus a worthwhile research goal. One possible direction towards reducing the optimization time is *learning* a mapping between LPEs and $p$-PEs.

**Future work directions:** Future work can target several directions. A first step might be to experiment with different formulations of the general positional encoding scheme defined in Equation (2). We would also like to test the effectiveness of our positional encodings on link prediction tasks, where LPE usually performs well. Finally, we want to find cases where $p$-eigenvectors are more expressive than usual positional encodings. A promising direction here might be the capability of $p$-eigenvectors with $p \to \infty$ to capture information on shortest-paths (Deidda et al., 2022).

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

## Appendix A. $p$-Laplacian

We start with the following definition of the relevant notions of the graph $p$-Laplacian and its eigenvectors:

**Definition 2 (p-Laplacian)** *Let $G$ be a graph with adjacency matrix $\mathbf{W} = (w_{i,j})_{i,j=1}^n$. For $p > 1$, the corresponding graph p-Laplacian $\Delta_p : \mathbb{R}^n \to \mathbb{R}^n$ operating on $f \in \mathbb{R}^n$ is defined by $(\Delta_p f)_i = \sum_i \phi_p(f_i - f_j)$, where $\phi_p(x) = |x|^{p-1}\text{sign}(x)$.*

For $p = 2$, the graph $p$-Laplacian $\Delta_p$ is the unnormalized graph Laplacian, which is the only linear graph $p$-Laplacian. Thus, a natural question arises: how eigenvectors or eigenvalues can be defined for general graph $p$-Laplacians.

**Definition 3** *Let $p > 1$. A non-zero vector $v$ is called a p-eigenvector of $\Delta_p$ if there exists $\lambda \in \mathbb{R}$ such that*

$$(\Delta_p v)_i = \lambda \phi_p(v_i)$$

*for all $i = 1, \ldots, n$. The value $\lambda$ is called the p-eigenvalue corresponding to $v$.*

The $p$-eigenvectors share many properties with the standard eigenvectors of the graph Laplacian. For instance, the connectivity of the graph is represented by the multiplicity of its first eigenvector. Previous works leveraged $p$-eigenvectors for graph representational learning, for example, Bühler and Hein (2009a), used the fact that for $p \to 1$ the $p$-eigenvectors converge to the ratio Cheeger cut. However, note that not all nice properties of the usual eigenvectors of the 2-Laplacian carry over. For example, some graphs have more eigenvectors than the number of vertices in the graph (Deidda et al., 2022). Let

$$Q_p(f) = \langle f, \Delta_p f \rangle = \frac{1}{2} \sum_{i,j=1}^n w_{ij} |f_i - f_j|^p$$

and

$$F_p(f) = \frac{Q_p(f)}{\|f\|} \, .$$

All $p$-eigenvectors can be characterized as stationary points of $F_p$, similarly to the standard Laplacian.

**Theorem 4 ((Bühler and Hein, 2009a))** *A vector $f$ is a critical point of $F_p(f)$ if and only if $f$ is a p-eigenvector.*

It is well-known that $p$-PEs are an approximation of $p$-eigenvectors (Bühler and Hein, 2009a). To better understand $p$-eigenvectors and $p$-PEs, we visualize the first three non-trivial $p$-PEs for $p = 1.2, 1.6$ and 2 of the Minnesota graph in Figure 2. Note that the case $p = 2$ is equivalent to the LPE, i.e., the eigenvectors of the unnormalized Laplacian.

## Appendix B. Proofs

**Lemma 5 (Bühler and Hein (2009a))** *Let $G$ be a graph with $k$ connected components. The set of p-LPEs corresponding to the p-eigenvalue 0 is a linear subspace with dimension $k$ spanned by the indicator functions of the connected components.*

**Lemma 6 (Bühler and Hein (2009b))** *Let $v$ be a p-eigenvector with nonzero p-eigenvalue. Then $\sum_i \phi(v_i) = 0$.*

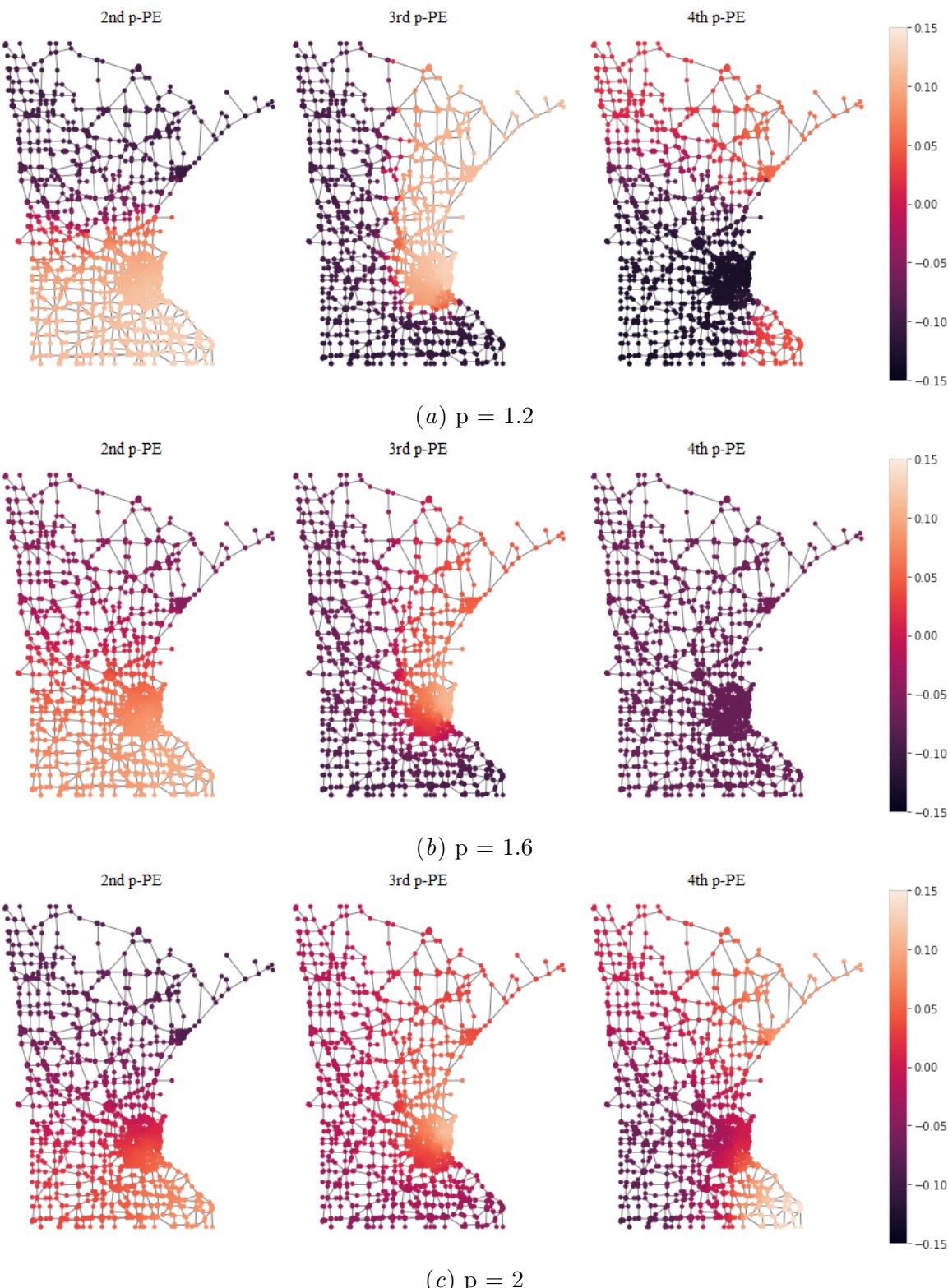

Figure 2: Comparison between $p$-PEs $(p = 1.2, 1.6, 2)$ on the Minnesota graph.

**Theorem 1** *For any $p \in (1, \infty)$, MPNNs augmented with at least two $p$-eigenvectors are strictly more expressive than the 1-WL test.*

**Proof** Let $G$ be a $k$-regular connected graph and $H$ a $k$-regular disconnected graph with the same number of nodes as $G$. For example, a 6-cycle as $G$ and two disjoint 3-cycles as $H$ suffices. 1-WL cannot distinguish $G$ and $H$ with the standard initial constant node features since these graphs are $k$-regular. Let $v^1, v^2$ be the given two $p$-eigenvectors for $G$ and $w^1, w^2$ the given two $p$-eigenvectors for $H$, sorted by their respective $p$-eigenvalues. By Lemma 5, $v^1$ is the constant vector on $V(G)$ and by Lemma 6, $v^2$ satisfies $\sum_i \phi(v_i^2) = 0$. For $H$, without loss of generality, we can assume that $w^1$ and $w^2$ are the indicator functions of the two different connected components of $H$ by Lemma 5. In particular, $\sum_i \phi(w_i^1) > 0$ and $\sum_i \phi(w_i^2) > 0$.

Let $(v_i^1, v_i^2)$ be the initial embedding of the $i$-th node in $G$ and $(w_i^1, w_i^2)$ be the initial embedding of the $i$th node of $H$. As MPNNs can learn the identify function (Morris et al., 2019), we can assume that message passing does not change these embeddings. Hence, we can use $v^2$ and $w^2$ to distinguish $G$ and $H$. We can apply a sufficiently large MLP (relying on the universal approximation theorem (Cybenko, 1989)) to approximate $\phi$ and then apply summation leading to a value that can distinguish the two cases $\sum_i \phi(w_i^2) = 0$ and $\sum_i \phi(v_i^2) > 0$. Finally, we claim that 1-WL is never more expressive as MPNNs with $p$-eigenvectors. For that, simply note that an MPNN can ignore the PE and run standard message passing. ∎

## Appendix C. Further Experimental Details

### C.1. Node-Classification Task

We consider the citation networks Cora and Citeseer (Lu and Getoor, 2003; Getoor et al., 2001) with the standard training, validation, and test splits, using only on the largest connected component in each network. As MPNN, we consider GCN (Kipf and Welling, 2017) with 2 layers and train with standard Adam. The tuning of the other hyperparameters was done using a standard random search algorithm, implemented via Weights&Biases (Biewald, 2020). The tuned hyperparamaters are the learning rate (0.01 - 0.1), the weight decay (0.001 - 0.1), the hidden channels $(8, 16, 32, 64)$, and the dropout (0 - 0.5). The hyperparameter configuration with the best validation accuracy is then trained over four different seeds. We report the mean accuracy and standard deviation over the test data set in Table 1.

### C.2. Graph-Regression Task

For the graph-regression experiments, we consider the ZINC dataset of molecule graphs (Irwin et al., 2012), where we only consider the subset of 12,000 graphs from (Dwivedi et al., 2020). We follow the implementation and configurations from (Lim et al., 2022), see Appendix J.2 and https://github.com/cptq/SignNet-BasisNet.

