# OpenReview forum: "Generalized Laplacian Positional Encoding for Graph Representation Learning"
_NeurIPS.cc/2022/Workshop/NeurReps — NeurReps 2022 Poster_

### Official Review · Reviewer_3NdT · 2022-10-10
**Review of Generalized Laplacian Positional Encoding for Graph Representation Learning**

**Confidence:** 4
**Soundness:** 3
**Presentation:** 3
**Contribution:** 2
**Overall Rating:** 6

**Summary:**

The paper proposes to consider more general "approximate" p-Laplacian eigenvectors as positional encoding rather than the classical Laplacian positional encoding (p = 2).

**Questions:**

See Limitations section above.

**Limitations:**

Yes.

**Recommended Decision:**

3: Accept

**Relevance:**

3: Solid fit

**Strengths And Weaknesses:**

Strengths: The idea is interesting to investigate and more specifically trying to assess the impact of different cost (d) and constraint (C) choices from a theoretical point of view in terms of expressivity and more specifically structures that can be recognised.

Limitations: As observed by the authors, the empirical results are not encouraging. However, that could be due to issues on the optimization level rather than on the usage of the generalized p-Laplacian eigenvectors. It also perhaps not extremely clear yet what choosing d and C differently from the LPE case might give us from a theoretical point of view; why would we expect this to be better beyond simply being more general?

**Submission Track:**

Extended Abstract (4 Page)

---

### Official Review · Reviewer_eT7q · 2022-10-14
**Unclear take-home message**

**Confidence:** 3
**Soundness:** 2
**Presentation:** 3
**Contribution:** 2
**Overall Rating:** 5

**Summary:**

The abstract builds on the ideas of enhancing GNNs with positional encodings. In particular, the authors propose a Laplacian-based positional encoding scheme that is based on generalising Laplacian embeddings, and Local Preserving Projections (LPPs) to p-norms.  These embeddings are used as an input to classical GNNs in order to improve their performance.

**Questions:**

More thorough and detailed experiments are needed. Also connections with kernel LPPs should be established.

**Limitations:**

Although the topic is promising and the approach valid, the results are not that encouraging. More detailed analysis should be done, probably starting with simpler/synthetic datasets.

**Recommended Decision:**

2: Borderline

**Relevance:**

3: Solid fit

**Strengths And Weaknesses:**

The work is technically sound, and given the space restrictions of an abstract, the claims are adequately discussed. The less encouraging parts of the abstract are the results who do not seem to confirm the initial intuition. Connections to kernels LPPs such as
Li, Jun-Bao, Pan, Jeng-Shyang, and Chu, Shu-Chuan. Kernel class-wise locality preserving projection. Information Sciences, 178(7):1825–1835, 2008
should be made.

**Submission Track:**

Extended Abstract (4 Page)

---

### Official Review · Reviewer_KeAd · 2022-10-15
**Interesting generelization but emperically LPE perform better than the proposed method**

**Confidence:** 3
**Soundness:** 3
**Presentation:** 3
**Contribution:** 2
**Overall Rating:** 5

**Summary:**

The paper generalizes the Laplacian-based positional encoding to improve expressive power and link predictions in GNNs. The work generalizes the LPEs expressed as solutions to the optimization problem of finding the embeddings that use 2-norm and have normalized orthogonality constraints. The generalized version uses $p$-norm on some dissimilarity function with the constraints $C(X)=0$. The paper provides the experimental results where they compare the performance of LPEs with $p$-PEs and find that the former performs better than the latter.

**Questions:**

1. From the experiments, it is clear that the LPE still performs better than the $p$-PEs. Could you reason for this behavior?
2. In which scenarios $p$-PEs should prove significant as opposed to LPEs?

**Limitations:**

As the authors have already stated, an important limitation is the performance of the $p$-PE that is outperformed by the LPEs. There could be significant directions that might have been overlooked where $p$-PE might prove efficient.

**Recommended Decision:**

2: Borderline

**Relevance:**

3: Solid fit

**Strengths And Weaknesses:**

The idea is interesting. The experimental results however do not support the expected behavior of the generalized positional encodings. Overall the paper is well-organized and written clearly with the challenges and limitations clearly stated.

**Submission Track:**

Extended Abstract (4 Page)

---

### Decision · Program_Chairs · 2022-10-21

Accept (Poster)